# MPCD Index for Hepatocellular Carcinoma Patients Based on Mitochondrial Function and Cell Death Patterns

**DOI:** 10.3390/ijms26010118

**Published:** 2024-12-26

**Authors:** Longxing Wang, Zhiming Zhao, Kunxian Shu, Mingyue Ma

**Affiliations:** 1Chongqing Key Laboratory of Big Data for Bio Intelligence, Chongqing University of Posts and Telecommunications, Chongqing 400065, China; s220502013@stu.cqupt.edu.cn (L.W.); s230502021@stu.cqupt.edu.cn (Z.Z.); shukx@cqupt.edu.cn (K.S.); 2College of Pharmacy, Chongqing Medical University, Chongqing 400016, China

**Keywords:** mitochondrial function, scRNA-seq, multi-omics, machine learning, staurosporine, prognosis

## Abstract

Hepatocellular carcinoma (HCC) is a highly heterogeneous cancer with a poor prognosis. During the development of cancer cells, mitochondria influence various cell death patterns by regulating metabolic pathways such as oxidative phosphorylation. However, the relationship between mitochondrial function and cell death patterns in HCC remains unclear. In this study, we used a comprehensive machine learning framework to construct a mitochondrial functional activity-associated programmed cell death index (MPCDI) based on scRNA-seq and RNA-seq data from TCGA, GEO, and ICGC datasets. The index signature was used to classify HCC patients, and studied the multi-omics features, immune microenvironment, and drug sensitivity of the subtypes. Finally, we constructed the MPCDI signature consisting of four genes (S100A9, FYN, LGALS3, and HMOX1), which was one of the independent risk factors for the prognosis of HCC patients. The HCC patients were divided into high- and low-MPCDI groups, and the immune status was different between the two groups. Patients with a high MPCDI had higher TIDE scores and poorer responses to immunotherapy, suggesting that high-MPCDI patients might not be suitable for immunotherapy. By analyzing the drug sensitivity data of CTRP, GDSC, and PRISM databases, it was found that staurosporine has potential therapeutic significance for patients with a high MPCDI. In summary, based on the characteristics of mitochondria function and PCD patterns, we used single-cell and transcriptome data to identify four genes and construct the MPCDI signature, which provided new perspectives and directions for the clinical diagnosis and personalized treatment of HCC patients.

## 1. Introduction

Hepatocellular carcinoma (HCC) originates from hepatocytes and is one of the major types of primary liver cancer, characterized by high incidence, delayed diagnosis, and poor prognosis [1]. In 2022, there were over 750,000 liver cancer deaths worldwide, positioning liver cancer as the third leading cause of cancer death and the sixth most frequently diagnosed [2]. Despite the long-standing identification and widespread awareness of major risk factors such as HBV or HCV chronic infection, aflatoxin exposure, heavy alcohol consumption, excess body weight, type 2 diabetes, and smoking, the HCC incidence rates in formerly low-risk countries, like those in Europe, North America, Australia/New Zealand, and South America, have increased. In China, HCC remains one of the top four causes of cancer death [3,4]. Inadequate early detection, poor risk stratification methods, and a lack of effective therapeutic options for patients with advanced stages all may contribute to the increased mortality of HCC [5]. The pathophysiology of HCC is multifaceted, driven by a complex interplay of genetic, epigenetic, and environmental factors [6]. The development of HCC is closely associated with a tumor-prone microenvironment characterized by persistent inflammation, extracellular matrix remodeling, and abnormal cellular regeneration. These processes result in genetic instability, the accumulation of somatic mutations, and epigenetic alterations, which collectively promote hepatocarcinogenesis [7]. At the molecular level, HCC is driven by genetic alterations, including mutations in key oncogenes and tumor suppressor genes such as *TERT*, *CTNNB1*, and *TP53* [8]. Additionally, the dysregulation of key signaling pathways, including the *Wnt*-*β*-catenin pathway, which is frequently activated in HCC and drives cell proliferation, and the *TGF-β* signaling pathway, known for its dual role in tumor suppression and progression, plays a critical role in hepatocarcinogenesis. The MAPK and *PI3K/AKT* pathways are also commonly altered, contributing to enhanced survival, growth, and metabolic reprogramming of HCC cells [9,10]. In recent years, with the rapid advancement of big data and biotechnology, some new therapeutic strategies combining chemotherapy drugs with immunotherapy have emerged [11]. However, the prognosis for patients remains poor, with a 5-year survival rate of less than 30% [12]. In addition, in up to 70% of HCC patients who undergo radical hepatectomy, HCC will recur within 5 years, which is very unfavorable for the long-term treatment of patients [13]. Therefore, establishing accurate molecular typing is crucial for revealing molecular mechanisms, early diagnosis, and personalized treatment strategies.

Cell death is a physiological process essential for maintaining biological development and homeostasis. Killing cancer cells on cell death-related pathways is one of the main research directions in cancer treatment. According to the triggering mechanisms, cell death is divided into two types: accidental cell death (ACD) and programmed cell death (PCD). ACD is a type of uncontrolled biological process, PCD is regulated by complex mechanisms [14]. PCD can be initiated by multiple pathways [15], and different types of cell death are not independent of each other; when one pathway is defective, other regulatory mechanisms will ensure the cell death process [16]. The different types of PCD can be viewed as a synergistic cell death system in which the pathways are closely correlated and flexibly complement each other [17]. PCD pathways drive the clearance of functionally redundant, infected, or potentially malignant cells, highlighting their critical role in maintaining homeostasis, resisting pathogens, and combating cancer, and other pathological conditions. In addition, certain PCD patterns can activate immune responses and stimulate powerful and sustained anti-tumor immunity. Therefore, by exploring potential targets related to the PCD and manipulating the PCD of tumor cells, the purpose of intervening or treating cancer can be achieved [18].

The dysfunction of many organelles is associated with the development and progression of cancer, especially mitochondria, which are metabolic centers for tumor cell proliferation, survival, and metastasis due to their important roles in bioenergetics and biosynthesis [19]. Mitochondrial dysfunction can induce cellular stress and initiate the PCD pathways [20]. The disruption of mitochondrial morphology can impair normal function and also induce PCD [21]. Recently, mitochondria have been widely considered to be one of the key drivers of inflammation associated with cell death [22]. Many studies have reported that in various types of liver diseases, mitochondria are regarded as multifunctional regulators of necroptotic mechanisms, capable of promoting inflammatory responses, enhancing immune reactions, and regulating disease progression [23,24,25]. An in-depth exploration of mitochondrial dysfunction and its mediated necroptotic activities could provide potential therapeutic targets for treating liver diseases. Mitochondrial dysfunction and PCD patterns are crucial for the development and spread of malignant tumors. Malignant cells must overcome various forms of cell death and mitochondrial dysfunction to promote malignant cell growth. However, a comprehensive understanding of the interaction between mitochondrial function and PCD patterns in HCC remains limited, with few detailed functional studies in this context. For example, Qin et al. [26] developed a prognostic model based on an inter-crosstalk definition of mitochondrial function and cell death patterns. In their study, only a simple association between genes related to mitochondrial function and PCD-related genes was established at the gene expression level (Pearson co-expression analysis, *r* > 0.6, *p* < 0.001). In general, the method was relatively simple and the data format is single. To fully understand the molecular mechanisms of mitochondrial dysfunction and cell death in tumors, information at the cellular level must be considered.

Single-cell sequencing technology provides unprecedented opportunities to understand the molecular characteristics of individual cells [27,28]. Previous studies have used single-cell RNA sequencing (scRNA-seq) data to identify immune phenotypes in the tumor microenvironment [29,30] and novel immune cell-associated functional biomarkers [31,32]. A comprehensive understanding of the molecular characteristics of a disease is crucial to the study of the disease. But, in the integrated use of scRNA-seq data and RNA-seq data, problems such as differences in data sources, batch effects, and technical bias are often encountered [33,34]. In our study, strict data quality control, data standardization conversion, canonical correlation analysis, and other methods were used to integrate different samples, reduce technical deviations between samples, and ensure data consistency.

Our study intends to use scRNA-seq data to mine genes related to mitochondrial function, and on this base, use RNA-seq to calculate the sample score as a key indicator of HCC. A new cell death signature was constructed based on a comprehensive machine learning framework, named the mitochondrial functional activity-associated programmed cell death index (MPCDI). HCC patients were divided according to the MPCDI signature, and the reliability of the new classification method was verified from multiple aspects, such as multi-omics characteristics, immune characteristics, and drug sensitivity.

## 2. Results

### 2.1. 1711 Mitochondrial Function Activity-Associated Genes Were Identified from HCC scRNA-Seq Data

Data from three HCC single-cell samples were downloaded from the GSE162616 dataset, and after quality control and integration filtering, a total of 38,311 cells were selected for subsequent analysis. The number of genes detected, the depth of sequencing, and the proportion of mitochondrial gene expression in three HCC single-cell samples are shown in Appendix A. There was a strong positive correlation between the number of detected genes and sequencing depth (*Pearson* test, *p* < 0.01, Appendix A). The optimal dimensionality reduction was determined to be 18 (Appendix A), the resolution was determined to be 0.3 (Appendix A), and the clusters were visualized using t-SNE (Figure 1A). The ROGUE analysis showed values greater than 0.85, indicating that the clusters were well-homogenized across the different samples (Appendix A). Using the Cell Taxonomy database, clusters were annotated into nine major cell types (Figure 1B,C), including NK cells identified by *NKG7* expression (*N* = 25,248), mast cells identified by *HPGDS* expression (*N* = 2877), T cells identified by *CD3D* expression (*N* = 6576), endothelial cells identified by *PECAM1* expression (*N* = 1467), plasma cells identified by *MZB1* expression (*N* = 994), B cells identified by *MS4A1* expression (*N* = 543), myeloid dendritic cells identified by *CLEC9A* expression (*N* = 269), macrophages identified by *C1QB* expression (*N* = 194), and hepatocytes identified by *ALB* expression (*N* = 143). We demonstrated the major cell subsets present in the tumor tissues of three patients and found that the degree infiltration of each subset varied (Figure 1D, Appendix A). AUCell analysis was used to assign mitochondrial functional activity scores to each cell subtype based on 1136 mitochondrial function-related genes, (Figure 1E and Appendix A). Based on median AUC scores, endothelial cells, hepatocytes, macrophages, myeloid dendritic cells, and plasma cells were classified into the high mitochondrial functional activity group, while B cells, mast cells, NK cells, and T cells were categorized into the low group (Figure 1F). Genes affecting mitochondrial functional activity were identified by a differential analysis of these two groups (*p* < 0.05). Finally, 1711 differentially active genes (DAGs) were obtained (Appendix A).

### 2.2. Thirty-Seven Key Genes Interacted with Mitochondrial Function and PCD Patterns

Based on the TCGA-LIHC dataset, samples were divided into patient and normal groups, and 4529 differentially expressed genes (DEGs, Figure 2A) were identified using the limma package (|log2FC| > 0.5 and FDR < 0.05). The expression matrices of DEGs were used for ssGSEA analysis to calculate the mitochondrial functional activity scores for each HCC patient. Subsequently, the expression matrices of PCD genes (*N* = 1796, and Appendix A) were utilized for WGCNA analysis, from which marker genes associated with mitochondrial functional activity scores were identified. The optimal soft threshold was seven, at which the scale-free topology index (*R*^2^) reached 0.9 (Figure 2B), and the 1796 PCD genes were divided into four modules (Figure 2C,D). Among four modules, the MEbrown had the highest correlation with mitochondrial functional activity scores (*Pearson* test, *cor* = 0.75, *p* = 4 × 10^−68^), containing 312 genes. The genes in the MEbrown module were also highly correlated with their corresponding mitochondrial functional activity scores, implying their important roles in mitochondrial function (Figure 2E). Then, the study considered the shared genes among three sources as the key genes, including differentially active genes from single-cell data analysis, differentially expressed genes from RNA-seq analysis, and the MEbrown module genes from WGCNA analysis. Ultimately, we identified 37 key genes involved in the interaction between mitochondrial function and PCD patterns (Figure 2F).

### 2.3. Construction and Evaluation of the MPCDI Signature

Combining clinical information from HCC patients, univariate Cox regression analysis was performed on 37 key genes, and 16 potential prognostic genes were identified (Appendix A). The TCGA-LIHC dataset was used as the training dataset, and the prognostic genes were incorporated into a comprehensive machine learning framework to construct a predictive model using 10-fold cross-validation. To ensure the applicability of the model, GSE14520, GSE116174, and ICGC datasets were used as validation datasets. The model with the highest average C-index across datasets was selected as the optimal model, and the prognostic model was finally identified as a combination of StepCox[both]+RSF (Figure 3A), and four signature genes (*S100A9*, *FYN*, *LGALS3*, and *HMOX1*) were identified. Then, a multivariate Cox regression analysis was performed to obtain the regression coefficients for each gene (Figure 3B). Based on the expression levels of these four genes and their corresponding regression coefficients, the MPCDI for each patient was calculated using the following formula: MPCDI = 0.1076824 × *S100A9* − 0.3186202 × *FYN* + 0.1077583 × *LGALS3* + 0.1628468 × *HMOX1*. Patients were divided into high- and low-MPCDI subtypes according to the optimal cutoff value (1.386066), and differences in overall survival (OS) between the two subtypes were compared. We found that patients with a high MPCDI had significantly lower survival rates in all four datasets, indicating a worse prognosis in the high-MPCDI group (Figure 3C–F). In addition, the MPCDI signature could well predict 1-, 3-, and 5-year OS (Figure 3G–J).

Clinicians commonly use clinical features (such as AJCC staging) and molecular characteristics (such as MSI and KRAS mutations) for prognosis assessment and management. Then, the prognostic value of the MPCDI signature and common clinical and molecular features were predicted in the four datasets (Appendix A). The C-indexes of the MPCDI signature were higher than that of almost all other features (age, gender, T, M, N, Stage, TMB, MSI, TP53 mutations, and KRAS mutations), indicating that the MPCDI signature had a good clinical application prospect. With the rapid development of next-generation sequencing, predictive features based on gene expression have been widely explored and developed. To further validate the reliability of the MPCDI signature, it was compared with 10 other gene expression-based prognostic signatures [35,36,37,38,39,40,41,42,43,44]. The comprehensive performance of the MPCDI signature on the four datasets was better than that of those ten signatures (Appendix A).

Then, univariate and multivariate Cox regression analyses were used to evaluate the significance of MPCDI as an independent prognostic factor. Both univariate Cox regression and multivariate analysis indicated that the MPCDI and stage were independent prognostic factors for HCC patients (Appendix A). To enhance the clinical applicability of the model, a nomogram was developed in the TCGA-LIHC cohort to estimate 1-, 3-, and 5-year OS using multivariate Cox and stepwise regression analysis. The C-index of the model was 0.658 (95% confidence interval: 0.604–0.713), including MPCDI and stage (Appendix A), and calibration curves demonstrated that the model had a high accuracy in predicting OS (Appendix A). Then, according to the median score of the Nomogram (Appendix A), there was a significant survival difference between the high and low groups (*p* < 10^−4^) (Appendix A). By calculating the AUC values of TCGA, GSE14520, GSE116174, and ICGC datasets, the results showed that the nomogram could accurately predict the 1-, 3-, and 5-year OS of HCC patients, with the average AUC value = 0.727 (Appendix A).

### 2.4. Multi-Omics Characteristics of High- and Low-MPCDI Subtypes

To explore molecular characteristics of high- and low-MPCDI subtypes, we performed multi-omics differential analysis between patients with the two MPCDI subtypes and normal samples, including mRNA, miRNA, lncRNA, and DNA methylation. A total of 1998 differentially expressed mRNA, 104 differentially expressed miRNA, 189 differentially expressed lncRNA, and 784 differential DNA methylation sites were found in the high-MPCDI subtype. A total of 1206 differentially expressed mRNA, 90 differentially expressed miRNA, 134 differentially expressed lncRNA, and 2125 differential DNA methylation sites were found in the low-MPCDI subtype (Figure 4A). There were 932 and 140 unique differential genes for high-MPCDI and low-MPCDI subtypes, respectively. The high-MPCDI subtype group also had 32 unique differentially expressed miRNAs, 70 unique differentially expressed lncRNA, and 237 unique differential DNA methylation sites. For the low-MPCDI subtype, there were 18, 15, and 1578, respectively.

Then, we explored the distribution of copy number variations (CNVs) and single nucleotide variations (SNVs) in the two subtypes. The frequency of recurrent copy number alterations was higher in the low-MPCDI subtype (Figure 4B,C). Additionally, different frequencies of gene-level mutations were associated with the types of MPCDI subtypes. In the high-MPCDI group, *PSMD4* exhibited the most significant copy number amplification and *FNDC3A* exhibited the most extensive copy number deletions (Appendix A). In the low-MPCDI group, *SQLE* exhibited the most significant copy number amplification, while *ANKRD65*, *ATAD3C*, *ERRFI1*, *GABRD*, and *TNFRSF4* exhibited the most extensive copy number deletions (Appendix A). The analysis of common SNVs in HCC patients revealed that approximately 84.83% (302/356) of patients exhibited mutations. Figure 4D displays the top 20 mutated genes, TP53 showing the highest mutation frequency (29%), and the mutation frequencies of other genes ranging from 6% to 26%. A comparison of differential mutation genes between the high- and low-MPCDI groups revealed that the mutation frequencies of *TP53* were lower in patients with a high MPCDI than in those with a low MPCDI (Figure 4D). Then, we explored the loci, CNVs, and SNVs of the four MPCDI signature genes (Figure 4E). The copies of *S100A9* (on Chr1) and *HMOX1* (on Chr22) were significantly amplified, and the copies of *FYN* (on Chr6) and *LGALS3* (on Chr14) were lost considerably. The mutations of *HMOX1* and *FYN* were single nucleotide polymorphisms (SNPs) and were missense mutations. Furthermore, *FYN* also exhibited a deletion mutation, specifically a frameshift deletion.

GSVA analysis showed that the high-MPCDI group was positively correlated with O-glycan biosynthesis, cellular response to leucine, a positive regulation of TORC1 signaling, viral protein processing, and lysosomes. In contrast, the low-MPCDI group was positively correlated with the intrinsic apoptotic signaling pathway, the negative regulation of macro-autophagy, chronic inflammatory response, and Toll-like receptor signaling (Appendix A). Additionally, 228 genes with significant differential expression between high- and low-MPCDI subtypes were identified. In the GSEA of Hallmark pathways, epithelial–mesenchymal transition, late estrogen response, and the complement system were extensively enriched in the high-MPCDI subtype. In addition, bile acid metabolism was more abundant in the low-MPCDI subtype (Appendix A).

This study suggested that the MPCDI signature might be a key microenvironmental factor for aggressive HCC and that the two MPCDI subtypes have distinct multi-omic features. Consequently, constructing an MPCDI subtype prediction model (Figure 5A) by integrating multi-omics features and deep neural networks helps predict the aggressiveness and prognosis of HCC patients. The predictors included in the deep learning model were the top 100 features with the greatest absolute fold changes in differentially expressed mRNA, lncRNA, miRNA, and methylation between high- and low-MPCDI subtypes. The monolayer-based predictor performed exceptionally well in predicting the MPCDI signature in training, testing, and entire datasets, indicating no obvious overfitting among the four MPCDI signature predictors (Figure 5B–D). In the training dataset, the AUCs of mRNA, miRNA, lncRNA, and DNA methylation were 0.951, 0.932, 0.965, and 0.955, respectively (Figure 5B). The GSE14520, GSE116174, and ICGC datasets were included as independent testing datasets to further explore the performance of the MPCDI signature predictors (Figure 5C). Since only mRNA data were available in these three testing datasets, we tested the mRNA-based MPCDI signature predictor and observed strong performance; the AUCs of GSE14520, GSE116174, and ICGC datasets were 0.967, 0.962, and 0.940, respectively (Figure 5E). Then, an interpretable deep neural network model (DNNCDF) was introduced to decode MPCDI features from multi-omics data. The DNNCDF outperformed all four single-layer MPCDI signature predictors, the AUCs of training, testing, and entire datasets were 0.973, 0.972, and 0.973, respectively (Figure 5F), highlighting the effectiveness of this strategy in combining multi-omics data. Patients were categorized into high- and low-MPCDI subtypes based on the DNNCDF model, and significant survival differences between the two subtypes were detected in the four datasets (*p* < 0.05 in all four datasets, Figure 5G–J). This suggested that DNNCDF could serve as a promising clinical predictive biomarker for precision treatment, facilitating rational targeted therapies in the future.

### 2.5. HCC Patients with High MPCDI Respond Poorly to Immunotherapy

To evaluate the role of the MPCDI signature in the HCC immune microenvironment, we investigated the relationship of the MPCDI signature with immune infiltrating cells and immune modulators, and found that the MPCDI signature was positively correlated with the majority of tumor immune infiltrating cells, calculated by the CIBERSORT algorithm, MCPcounter algorithm, and TIMER algorithm. The MPCDI signature was positively correlated with ImmuneScore and ESTIMATEScore, and negatively correlated with tumor purity (Figure 6A). Additionally, the MPCDI signature was positively correlated with most of the immune modulators (genes) such as antigen presentation, cell adhesion, co-inhibitors, co-stimulators, ligands, and receptors (Figure 6B). We also examined the utility of the MPCDI signature in predicting patients’ benefit from immune therapy. Patients in the low-MPCDI group in the SKCM cohort had a significantly longer survival (Figure 6C) and a better response to anti-PD-1 therapy (Figure 6D) than those in the high-MPCDI group. These findings suggested that the MPCDI signature might be associated with immunotherapy response. The TIDE algorithm combined with the MPCDI signature was used to evaluate the potential clinical effect of immune therapy. A higher TIDE prediction score indicated a greater likelihood of immune escape and a lower probability of benefiting from immune therapy. The high-MPCDI subtype had higher TIDE scores, and TIDE scores were significantly positively correlated with the MPCDI signature (Figure 6E,F). This indicated that patients with high MPCDI may have poorer responses to immune therapy.

### 2.6. Staurosporine Was a Potential Drug for HCC Patients with High MPCDI

The CTRP, GDSC, and PRISM datasets were used to construct drug response prediction models, which contain gene expression profiles and drug sensitivity profiles for hundreds of cancer cell lines. After data filtering, 829 cancer cell lines (containing 408 compounds) from the CTRP dataset, 805 cancer cell lines (containing 179 compounds) from the GDSC dataset, and 558 cancer cell lines (containing 4659 compounds) from the PRISM dataset were obtained for further analysis. Two indicators were considered in identifying drug candidates with higher drug sensitivity in high-MPCDI patients. On the one hand, differential drug response analysis was performed between patients with high and low MPCDI to identify compounds with lower estimated scores in the high-MPCDI subtype (log2FC < −0.3). On the other hand, the *Spearman* test was used to calculate the correlation between the estimated scores and the MPCDI signature, and compounds with negative correlation coefficients (Spearman’s *r* < −0.3) remained. Finally, 14 CTRP-derived compounds, five GDSC-derived compounds, and nine PRISM-derived compounds were identified. In the high-MPCDI subtype, the estimated scores of all of these compounds were lower (Figure 7A,C,E) and negatively correlated with the MPCDI signature (Figure 7B,D,F). Then, CMap analysis was used to explore the therapeutic potential of these compounds in HCC, including the induction of gene expression patterns opposite to those specific to HCC, i.e., gene expressions that were elevated in tumor tissues but decreased under treatment with certain compounds. Only one compound, staurosporine, had a CMap score below −80 (Figure 7G). It suggested that staurosporine might have a potential therapeutic role in HCC patients with a high MPCDI.

### 2.7. Expression Distribution of the Four MPCDI Signature Genes

This study also calculated the gene expression and protein expression levels of four MPCDI signature genes, *S100A9*, *FYN*, *HMOX1*, and *LGALS3*. The gene expression data were from the TCGA-HCC and the GSE14520 dataset, and the protein expression data were from the CPTAC dataset of the UALCAN database. The gene expression and protein expression levels of the four genes were consistent (Figure 8A,B and Appendix A). Compared with normal samples, the expressions of *S100A9*, *FYN*, and *HMOX1* were downregulated, while *LGALS3* was upregulated in HCC patients. The immunohistochemical analysis of the HPA database also confirmed that the protein expression of *FYN* and *HMOX1* were downregulated, and that of *LGALS3* was upregulated in HCC (Figure 8C).

## 3. Discussion

HCC is one of the diseases that seriously endangers human health. Understanding its molecular mechanisms is crucial for clinical diagnosis and treatment. Studies have shown that the molecular mechanisms associated with HCC progression include epithelial–mesenchymal transition, tumor–stromal interactions, tumor microenvironment, senescence bypass, and immune regulation, closely related to programmed cell death patterns and mitochondrial dysfunction [45]. With the rapid advancement of computational biology, machine learning methods have been widely used to analyze biological big data. However, effectively implementing these methods in clinical practice and maintaining accuracy remains challenging. In this process, two key issues merit careful consideration: first, determining which specific machine learning algorithm is appropriate, and second, identifying the optimal solution. Researchers’ choices of algorithms may heavily rely on their preferences and biases [46].

This study used a comprehensive machine learning framework based on scRNA-seq data and RNA-seq data to identify the mitochondrial functional activity-associated programmed cell death genes. Firstly, 1711 genes related to mitochondrial functional activities were identified using HCC scRNA-Seq data. Secondly, 37 key genes interacting with mitochondrial function and their PCD patterns were obtained through WGCAN analysis. Then, 16 potential prognostic genes were found through univariate Cox risk regression. Next, the comprehensive machine learning framework was used to build a prediction model, and the most suitable model, StepCox[both]+RSF, was found from 101 combinations of machine learning algorithms (Figure 3A). The model consisted of four mitochondrial functional activity-associated programmed cell death genes, *S100A9*, *FYN*, *LGALS3*, and *HMOX1*. The gene expression levels of the four genes multiplied by the corresponding regression coefficients were calculated as the MPCDI signature. In all of the training and testing datasets, the MPCDI signature had a high accuracy in predicting 1-, 3-, and 5-year OS (Figure 3G–J). The comprehensive performance of the MPCDI signature also outperformed the common clinical features and the other ten gene-based signatures (Appendix A). Additionally, both univariate and multivariate Cox regression analyses suggested that the MPCDI signature was an independent risk factor affecting patient prognosis (Appendix A).

According to the MPCD index, HCC patients were divided into the high- and low-MPCDI groups. The high-MPCDI subtype group had 932 unique differentially expressed genes, 32 unique differentially expressed miRNAs, 70 unique differentially expressed lncRNA, and 237 unique differential DNA methylation sites. For the low-MPCDI subtype, there were 140, 18, 15, and 1578, respectively (Figure 4A). There are also obvious differences in the distribution of CNVs and SNVs between the two subtypes (Figure 4B,C). The OS rates in the high-MPCDI group were significantly lower than those in the low-MPCDI group (Figure 3C–F). In addition, the tumor immune microenvironment is closely associated with the prognosis of HCC and the efficacy of immunotherapy [47]. The high-MPCDI subtype patients had richer immune cell infiltration and higher expression levels of immune checkpoint molecules. However, patients with a high MPCDI had lower immune scores and ESTIMATE scores (Figure 6A), indicating a complex interaction between the MPCDI signature and the tumor immune microenvironment. Then, due to the absence of public datasets on immune-related treatments for HCC patients, we used the external immunotherapy datasets to validate the predictive ability of the MPCDI signature for immunotherapy response. The results indicated that patients with a high MPCDI had shorter survival times (Figure 6C), poorer responses to anti-PD-1 therapy (Figure 6D), and higher TIDE scores (Figure 6E). So, we believed that the MPCDI signature could serve as a predictive biomarker for immunotherapy responses, and patients with a high MPCDI were not suitable for immunotherapy.

Currently, targeted therapy and immunotherapy are developing rapidly [48]. During the treatment, HCC patients often develop drug resistance [49]. Sorafenib and lenvatinib are commonly used to treat advanced HCC, but their effectiveness only lasts about three months [50,51]. Developing more targeted drugs for specific populations is an urgent issue to be solved [52]. By analyzing the information from the CTRP, GDSC, and PRISM datasets, we found that staurosporine was a potential therapeutic drug for patients with a high MPCDI (Figure 7). Staurosporine is a multi-kinase inhibitor with various mechanisms for inducing apoptosis, and mainly through the mitochondrial pathway [53]. When HCC cells are co-cultured with tetralinoleoyl cardiolipin and treated with staurosporine, there is a significant restoration of apoptosis sensitivity, accompanied by an increase in cardiolipin and its oxidation products [54]. In addition, bisindolylmaleimide is a derivative of the PKC inhibitor staurosporine, which has shown potential as an anticancer drug in clinical trials [55]. For example, in both in vitro and in vivo experiments, bisindolylmaleimide alkaloid 155CI (BMA-155CI) significantly affected autophagy and apoptosis in human HCC HepG-2 cells [56]. Therefore, we believe that staurosporine can provide a better therapeutic effect on HCC patients with a high MPCDI.

The above analysis demonstrated the reliability of the MPCD index for HCC classification, which was composed of four genes related to the regulation of mitochondrial function, namely *S100A9*, *FYN*, *LGALS3*, and *HMOX1*. *S100A9* (S100 calcium-binding protein A9) is a member of the S100 protein family and plays an important role during the occurrence, progression, and metastasis of tumors, and is a very valuable cancer biomarker or novel therapeutic target [57,58]. *S100A9* may prevent cell death by activating reactive oxygen species (ROS)-dependent signaling pathways [59]. Studies have shown that low expression of *S100A9* is associated with longer survival [60]. The results of this study were consistent with this; HCC patients with a low MPCDI had longer survival and low *S100A9* expression. *FYN* is a non-receptor tyrosine kinase [61] that induces epithelial–mesenchymal transition, inhibits anti-tumor immune responses, and promotes tumorigenesis and metastasis [62]. Lin et al. found that *FYN* was a protective factor in HCC, with a negative coefficient in the risk function [63]. This was consistent with this study. *LGALS3* (galectin-3) is a galactoside-specific lectin that plays a key role in tumor microenvironment immunosuppression and regulates multiple cellular functions involved in cancer biology and cellular homeostasis [64]. *LGALS3* has been reported to be highly expressed in various cancers, and negatively correlated with the prognosis of HCC patients [65]. Bhat et al. found that the upregulation of *LGALS3* is significantly associated with recurrence in HCC [66]. Zhang et al. discovered that *LGALS3* was one of the key genes for bone metastases and related complications in HCC [67]. Similarly, we also found that *LGALS3* was a risk factor for HCC patients. *HMOX1* (Heme oxygenase 1) is an inducible enzyme [68] that plays an important role in maintaining cellular homeostasis and regulating critical biological processes such as oxidative stress, inflammation, cell proliferation, therapeutic resistance, and poor prognosis [69,70]. Accumulating evidence indicated that *HMOX1* exerts cytotoxic effects when its intracellular expression level exceeds a certain threshold [71,72]. Zheng et al. found that donafenib and GSK-J4 synergistically enhance the expression of *HMOX1*, increase intracellular Fe2+ levels, and ultimately lead to ferroptosis [73]. In our study, a high expression of *HMOX1* was not conducive to the prognosis of HCC patients.

In summary, the four genes related to mitochondrial function regulation and programmed cell death play critical roles in the development and progression of HCC and were potential biomarkers for HCC. In addition, the MPCDI signature was a reliable indicator for HCC patient classification and was helpful for clinical diagnosis and treatment. This work still needs some in-depth verification in the future. Firstly, due to the limitations and lags of current public data sets, the MPCD index still needs to be validated in more appropriate multicenter cohorts. Secondly, the four potential target genes and the therapeutic effects of staurosporine on patients with a high MPCDI require more direct experimental evidence.

## 4. Materials and Methods

### 4.1. Data Acquisition and Pre-Processing

Clinical information and gene expression profiles of HCC samples were downloaded from The Cancer Genome Atlas (TCGA, https://portal.gdc.cancer.gov/, accessed on 7 November 2023), Gene Expression Omnibus (GEO, https://www.ncbi.nlm.nih.gov/gds, accessed on 7 November 2023), and International Cancer Genome Consortium (ICGC, https://dcc.icgc.org/, accessed on 7 November 2023) databases. Samples without survival time and status were excluded, and only samples with survival times greater than 0 days were retained (Table 1). Gene expression levels were presented in the transcripts per million (TPM), and preprocessed by log2(X + 1) transformation. The scRNA-seq data were from three HCC samples (GSM4955419, GSM4955420, and GSM4955421) from the GSE162616 dataset [74].

Genes (*N* = 1136) associated with mitochondrial function (Appendix A) were extracted from MitoCarta3.0. 20 patterns of PCD and their key regulatory genes were collected through the literature search [26,75,76], including 580 genes related to apoptosis, 52 related to pyroptosis, 88 related to ferroptosis, 367 related to autophagy, 101 related to necroptosis, 19 related to cuproptosis, 9 related to parthanatos, 15 related to entotic cell death, 8 related to netotic cell death, 220 related to lysosome-dependent cell death, 7 related to alkaliptosis, 5 related to oxeiptosis, 24 related to NETosis, 34 related to immunogenic cell death, 338 related to anoikis, 66 related to paraptosis, 8 related to methuosis, 23 related to entosis, 15 related to disulfidptosis, and 485 related to PANoptosis. There were 2464 PCD-related genes, and we deleted 668 duplicates, resulting in 1796 PCD-related genes for analysis (Appendix A).

### 4.2. Single-Cell RNA-Seq Data Analysis

The “Seurat” package (version 5.0.3) was used to convert raw scRNAseq data to Seurat format and normalized by the LogNormalize method. scRNA-seq data quality control was performed by excluding genes expressed in fewer than three single cells, cells expressing fewer than 200 or more than 3000 genes, and cells with mitochondrial gene proportions exceeding 10%. The “FindVariableFeatures” function was used to identify the top 2000 highly variable genes. Then, the samples were integrated and the CCA method was used to eliminate the batch effects. All genes were scaled using the “ScaleData” function. Principal component analysis (PCA) was used to reduce the data dimensionality of the highly variable genes using the “RunPCA” function. The “FindNeighbors” and “FindClusters” functions were used to cluster cells. The “ElbowPlot” function was used to select the optimal dimensionality reduction. The “clustree” package (version 0.5.1) was employed to determine the appropriate resolution. Clusters were visualized using the t-distributed stochastic neighbor embedding (t-SNE) algorithm. The cluster purity was calculated through the ROGUE method [77]. The Cell Taxonomy database was used for manual cell annotation. The AUCell analysis [78] was used to determine the activity status of mitochondrial gene sets in the scRNA-seq data and calculate the mitochondrial functional activity scores for each cell subset. The subgroups were divided into high- and low-mitochondrial functional activity groups according to the median AUC score, for subsequent analysis of differentially active genes between the high and low groups.

### 4.3. Construction of Weighted Gene Co-Expression Network

The single sample gene set enrichment analysis (ssGSEA) [79] algorithm was used to quantify the relative abundance of each gene. In this study, the ssGSEA method was used to calculate the mitochondrial functional activity scores of each tumor sample based on the differentially active genes. The weighted gene co-expression network analysis (WGCNA) [80] was used to investigate PCD-related genes associated with mitochondrial functional activity scores. A soft threshold was selected when the scale independence reached 0.9, with 50 set as the minimum module size. Pearson correlation analysis was used to assess the correlation between module eigengenes and mitochondrial functional activity scores. Considering both coefficients and *p* values, the optimal module was selected.

### 4.4. Development of Prognostic Model

Univariate Cox regression analysis was performed on the key genes in HCC patient samples, and prognostic genes were screened out with *p* < 0.05. In the training set (TCGA-LIHC), the initial gene signatures were constructed using a comprehensive machine learning framework comprising 10 machine learning algorithms: Random Survival Forest (RSF), Elastic Net (Enet), Lasso, Ridge, Stepwise Cox, Cox Boost, Cox Partial Least Squares Regression (plsRcox), Supervised Principal Component (SuperPC), Generalized Boosted Models (GBM), and Survival Support Vector Machine (survival-svm) [46,81]. The RSF was implemented by the randomForest package (version 4.7.1.1) with the parameters: ntree = 1000 and nodesize = 5. The Enet, Lasso, and Ridge were implemented by the glmnet package (version 4.1.8), the regularization parameter λ was determined by cross-validation, and the L1-L2 trade-off parameter α was set to 0–1 (interval = 0.1). The stepwise Cox was implemented by the survival package (version 3.5.8), using the stepwise algorithm of AIC (Akaike information criterion), with the directional modes of the stepwise search set to “both”, “backward”, and “forward”, respectively. The CoxBoost was implemented by the CoxBoost package (version 1.5). The plsRcox was implemented by the plsRcox package (version 1.7.7), the cv2.plsRcox function was used to determine the required number of components, the plsRcox function was used to fit generalized linear models for partial least squares regression. The SuperPC was implemented by the superpc package (version 1.12), and it was a supervised PCA. The GBM was implemented through the gbm package (version 2.1.9), cross-validation was used to select the number of decision trees with the smallest cross-validation error, and the gbm function was used to fit the generalized boosting regression model. The survive-SVM was implemented through the survivalsvm package (version 0.0.5); this algorithm combined the classification ability of support vector machines and the characteristics of survival analysis. Based on 10-fold cross-validation, the prognostic genes were incorporated into 101 algorithm combinations, and further validated in testing datasets (GSE14520, GSE116174, and ICGC). The concordance index (C-index) was used to evaluate the predictive capability of the models. For each model, the C-index of each dataset (training set and testing sets) was calculated, and the model with the highest average C-index was selected as the optimal model. The optimal model was used to calculate the MPCDI. The “survminer” package (version 0.4.9) was used to determine the optimal cutoff value to classify patients into high- and low-MPCDI subtypes. The prognostic significance of the two subtypes was evaluated using Kaplan–Meier curves, and the prognostic efficacy of the MPCDI signature was assessed using receiver operating characteristic (ROC) curves.

### 4.5. Construction and Evaluation of Nomogram

The univariate and multivariate Cox regressions were used to analyze clinical characteristics (age, gender, and stage) of the two MPCDI subtypes. Prognostic nomograms were constructed through multivariate Cox and stepwise regression analyses, and the “regplot” package (version 1.1) was used to plot the nomograms. The prognostic performance of the nomogram was evaluated using Kaplan–Meier curves, calibration curves, decision curve analysis (DCA), and ROC curves.

### 4.6. Multi-Omics Characteristics of HCC Subtypes

Using |log2FC| > 1 and FDR (false discovery rate) <0.05 as the criterion, differentially expressed mRNA, miRNA, and lncRNA between HCC subtypes and normal samples were identified. For differential methylations, a criterion of |log2FC| > 0.4 and FDR < 0.05 was used. The GISTIC2.0 was used to analyze the copy number variation (CNV) between the two subtypes. The “maftools” package (version 2.18.0) was used to examine gene mutations. Some features, such as MSI and KRAS mutation, were only weakly associated with HCC, and were therefore not included in subsequent analyses. The gene set variation analysis (GSVA) was used to investigate the biological mechanisms of the specific genes of the subtypes. The Gene Set Enrichment Analysis (GSEA) was used to compare differences in Hallmark pathway enrichment between the two subtypes.

To decode the MPCDI microenvironment by combining multi-omics features, the TCGA dataset was divided into training and testing sets in a 1:1 ratio, with high-MPCDI and low-MPCDI patients equally distributed in each dataset. Using FDR < 0.05 as the criterion, the differential expression matrices of mRNA, miRNA, lncRNA, and methylation data between the two subtypes were analyzed. For each data type, the top 100 most relevant features were retained as markers associated with HCC-specific cell death. Subsequently, an artificial neural network with stacked autoencoders was used to learn the feature structure, reducing the number of marker features to 20 for each type of data [82]. Using deep neural networks, predictors of cell death features were constructed from 20 markers derived from each molecular layer. Ultimately, a deep neural network of cell death feature (DNNCDF) model was trained based on 80 specific cell death markers from four types of data. This model was designed to decode the MPCDI signature and distinguish between the two HCC subtypes.

### 4.7. Immune Microenvironment Analysis

The degree of immune infiltration was assessed by Cell-type Identification by Estimating Relative Subpopulations of RNA Transcripts (CIBERSORT) [83], Microenvironment Cell Populations-counter (MCPcounter) [84], and Tumor Immune Estimation Resource (TIMER) [85]. Each algorithm estimates the abundance of immune cell subsets through different strategies and gene expression signatures. Additionally, STromal and Immune cells in MAlignant Tumor tissues were estimated using the expression data (ESTIMATE) algorithm [86] to generate an overall immunological score. We also compared the correlations of the MPCDI signature with immune modulators (genes), including antigen presentation, cell adhesion, co-inhibitors, co-stimulators, ligands, and receptors [87]. An independent SKCM dataset was used to evaluate the predictive value of the MPCDI signature in immune therapy responses. The SKCM dataset was obtained from the GSE78220 project [88], and complete clinical data for 27 melanoma patients who received immunotherapy were obtained. The Tumor Immune Dysfunction and Exclusion (TIDE) algorithm [89] was used to predict immune therapy responses in HCC subtypes.

### 4.8. Drug Sensitivity Analysis

Drug sensitivity data were derived from the Genomics of Drug Sensitivity in Cancer (GDSC), the Cancer Therapeutics Response Portal (CTRP), and the PRISM database. The half-maximal inhibitory concentration (IC50) was used to quantify drug sensitivity. Compounds with more than 20% missing data were excluded, and the missing values were estimated using the k-nearest neighbors (k-NN) method [52]. Using the “oncoPredict” package (version 0.2), drug responses in clinical samples were predicted based on gene expression profiles, and drug susceptibility scores for each compound in each clinical sample were estimated. Lower estimated scores indicate greater sensitivity to treatment. Drug response data from CTRP, GDSC, and PRISM were compared to find candidate drugs with higher drug sensitivity in patients. Finally, CMap analysis was used to further validate and screen candidate drugs (scores < −95).

### 4.9. The Protein Expression Level of the Target Genes

UALCAN (https://ualcan.path.uab.edu/index.html, accessed on 3 July 2024) [90] is a comprehensive, user-friendly, and interactive web resource designed for analyzing cancer omics data. The protein expression (*z*-value) of MPCDI signature genes was obtained from the CPTAC dataset of the UALCAN database. The Human Protein Atlas (HPA, https://www.proteinatlas.org/, accessed on 20 March 2024) [91] utilizes transcriptomics and proteomics technologies to study protein expression at the RNA and protein levels across various tissues and organs. We used the HPA database to verify the protein expression levels of target genes in HCC tumor tissues and normal liver tissues.

### 4.10. Statistical Analysis

This study utilized R (version 4.3.1) for data processing, statistical analysis, model building, and data visualization, and Python (version 3.9) for the construction of a deep neural network. The Wilcoxon test was employed to compare the differences. The Pearson or Spearman correlation tests were employed for correlation analysis. Survival curves were depicted using Kaplan–Meier plots and compared using the log-rank test. A two-sided *p* value of less than 0.05 was considered statistically significant.

## 5. Conclusions

This study integrated HCC scRNA-seq and RNA-seq data and developed a tumor classification signature (MPCD index) based on extensive bioinformatics analysis and machine learning algorithms, and found four prognostic genes for HCC, *S100A9*, *FYN*, *LGALS3*, and *HMOX1*. The stability and reliability of the MPCDI signature were verified in multiple datasets, and the signature was one of the independent prognostic factors for HCC. Patients with a high MPCDI had poor survival and were insensitive to immunotherapy, and staurosporine may be more suitable for the high-MPCDI subtype. In summary, the MPCD index helps predict the aggressiveness and prognosis of HCC, and provides a new perspective for the study of the personalized treatment of HCC.

## Figures and Tables

**Figure 1 ijms-26-00118-f001:**
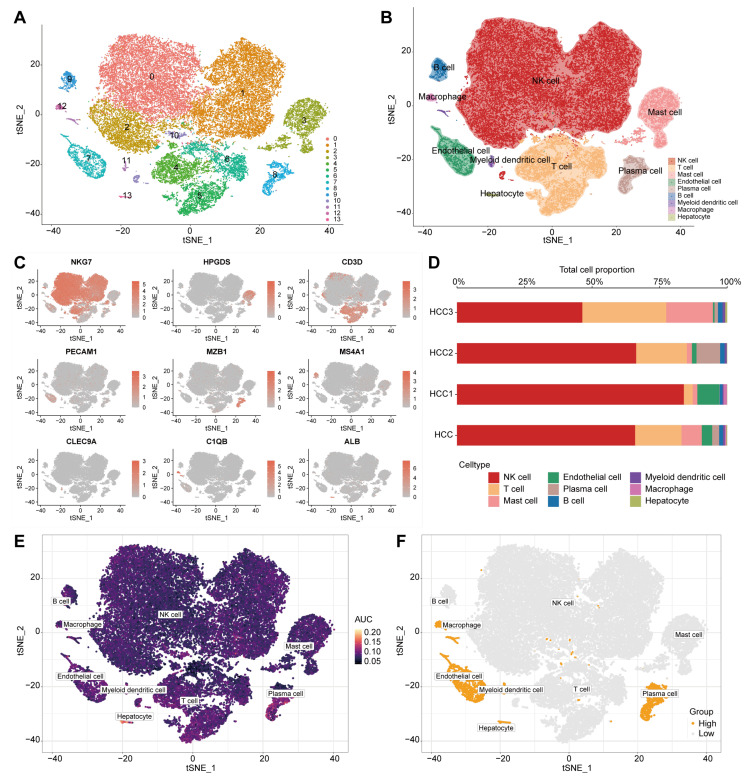
Single-cell atlas of tumor tissue from HCC patients. (**A**) Visualization of 14 clusters using the t-SNE algorithm. (**B**) Cell subsets annotated by marker genes. (**C**) Expression levels of marker genes. (**D**) Proportional distribution chart of nine cell types in different HCC patients and integrated samples. (**E**) Mitochondrial functional activity scores for each cell subset. (**F**) High and low mitochondrial functional activity groups.

**Figure 2 ijms-26-00118-f002:**
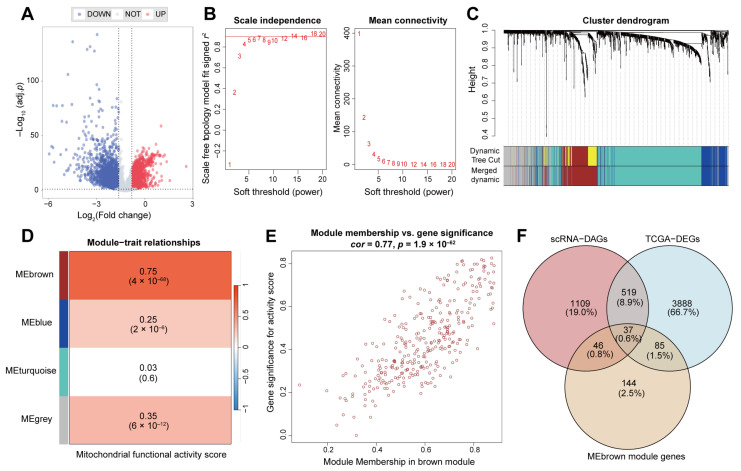
Identification of key genes for the interaction between mitochondrial function and PCD patterns. (**A**) Differentially expressed genes between TCGA-LIHC patients and normal samples. (**B**) Determination of the optimal soft threshold. (**C**) Clusters dendrogram. (**D**) The correlation between modules and mitochondrial functional activity scores. (**E**) Correlations of genes within the MEbrown and mitochondrial functional activity scores. (**F**) Number of overlapping genes among scRNA-DAGs, TCGA-DEGs, and MEbrown module genes.

**Figure 3 ijms-26-00118-f003:**
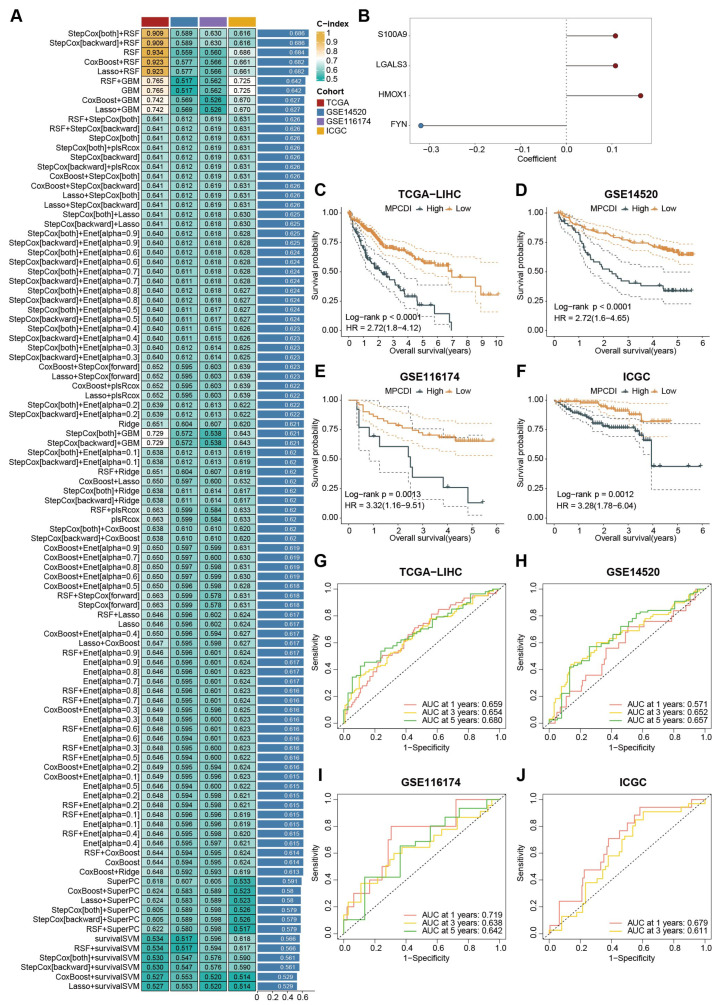
Prognostic value of the MPCDI signature. (**A**) C-index of 101 prediction models in four datasets. (**B**) Regression coefficients of the four signature genes. (**C**–**F**) Survival analysis between high- and low-MPCDI subtypes in the four datasets. (**G**–**J**) ROC curves of 1-, 3-, and 5-year OS for the four datasets.

**Figure 4 ijms-26-00118-f004:**
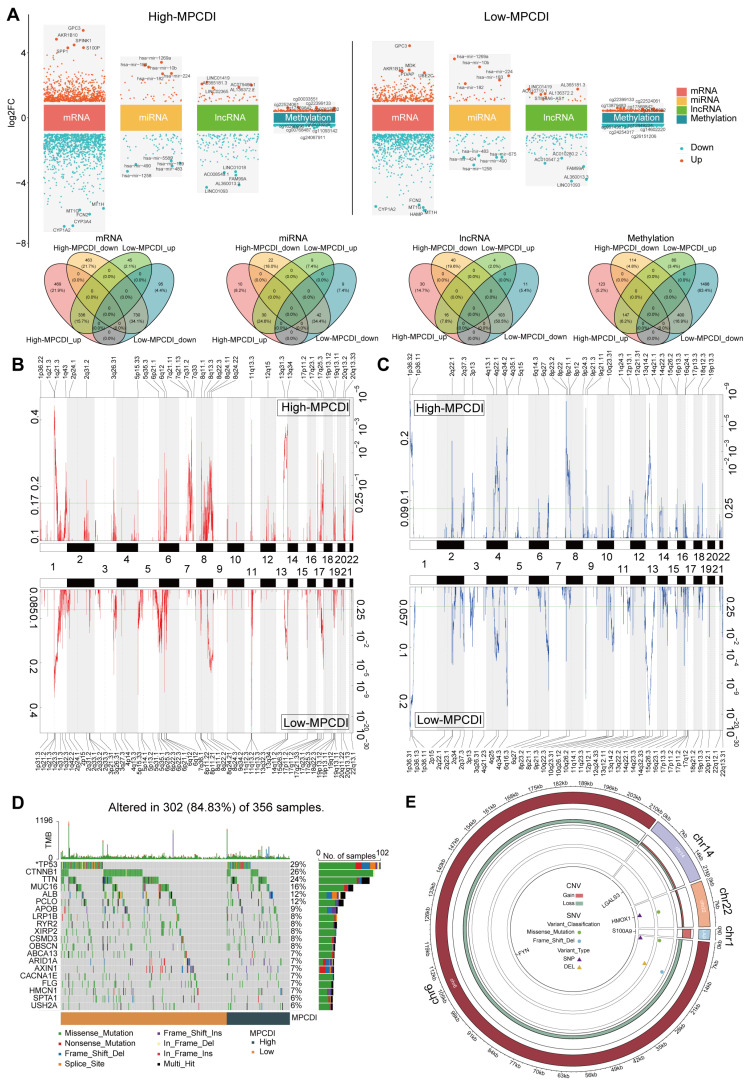
Single-cell atlas of tumor tissue from HCC patients. (**A**) Differential analysis between tumor and normal samples in mRNA, miRNA, lncRNA, and DNA methylation of high- and low-MPCDI subtypes, highlighting the top five positive and negative features for each molecular layer. (**B**) The copy number amplification of the two subtypes. (**C**) The deletion of the two subtypes. (**D**) Waterfall plot illustrating common somatic gene mutations, with the bar graph on the right depicting the corresponding mutation rates in each group. (**E**) Loci, CNV, and SNV information for MPCDI signature genes. * *p* < 0.05.

**Figure 5 ijms-26-00118-f005:**
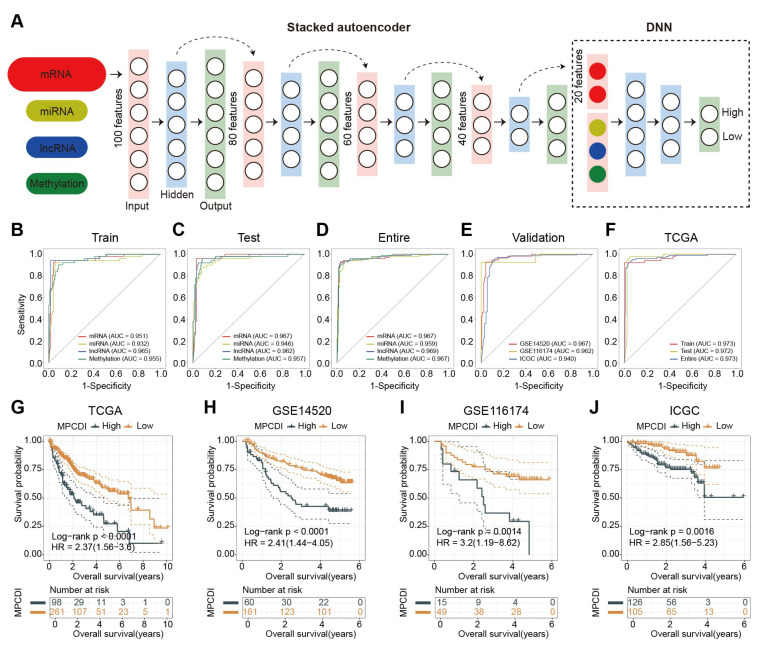
Performance of the deep neural network. (**A**) Workflow for integrating multi-omics features based on the DNNCDF. The TCGA dataset was divided into a training dataset (180 HCC tumor samples) and a testing dataset (179 HCC tumor samples). A four-level stacked autoencoder was used to learn feature structures from the training dataset, reducing the number of markers per molecular layer to 20. Based on 80 HCC-specific markers from four molecular layers, the DNNCDF was developed to decode the MPCDI signature and distinguish between different MPCDI subtype tumors. (**B**–**E**) ROC curves for the MPCDI signature predictors based on single molecular layers. (**F**) ROC curve for the DNNCDF model applied to the TCGA dataset. (**G**–**J**) Kaplan–Meier curves for OS of high- and low-MPCDI subtypes predicted by DNNCDF.

**Figure 6 ijms-26-00118-f006:**
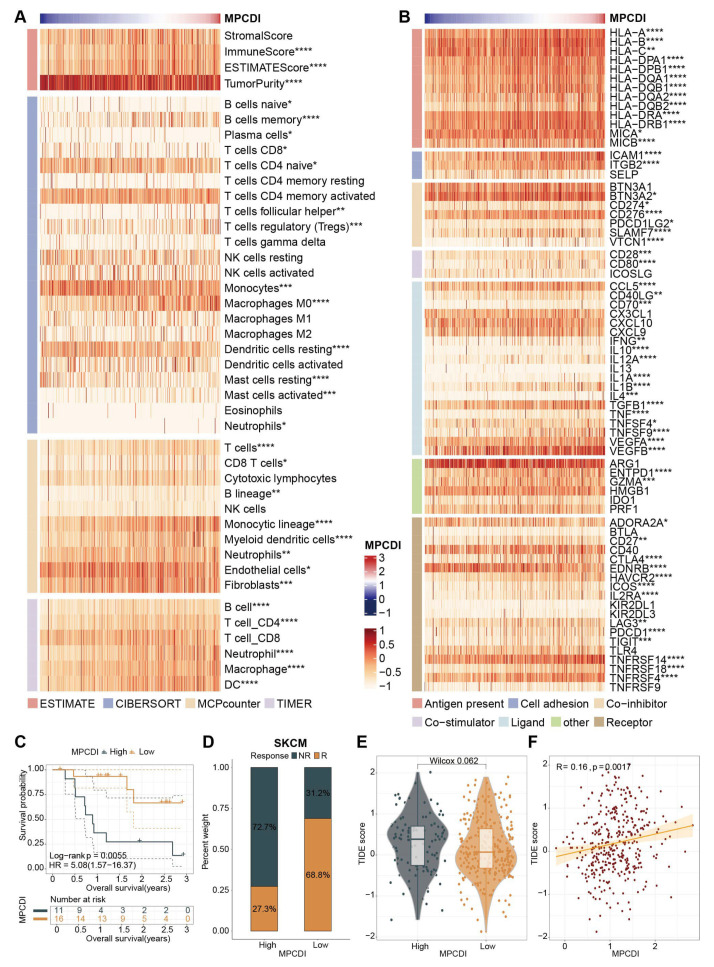
Immune-related characteristics of the MPCDI signature. (**A**) The correlations between the MPCDI signature and immune infiltrating cells, by the ESTIMATE algorithm. (**B**) The correlations between the MPCDI signature and immune modulators (genes). (**C**) Kaplan–Meier curves for high- and low-MPCDI subtypes in the SKCM cohort. (**D**) Reactivity to anti-PD-1 therapy in high- and low-MPCDI subtypes in the SKCM cohort. (**E**) Comparison of TIDE scores between high- and low-MPCDI subtypes. (**F**) Correlation between the MPCDI signature and TIDE scores. **** *p* < 0.0001, *** *p* < 0.001, ** *p* < 0.01, and * *p* < 0.05.

**Figure 7 ijms-26-00118-f007:**
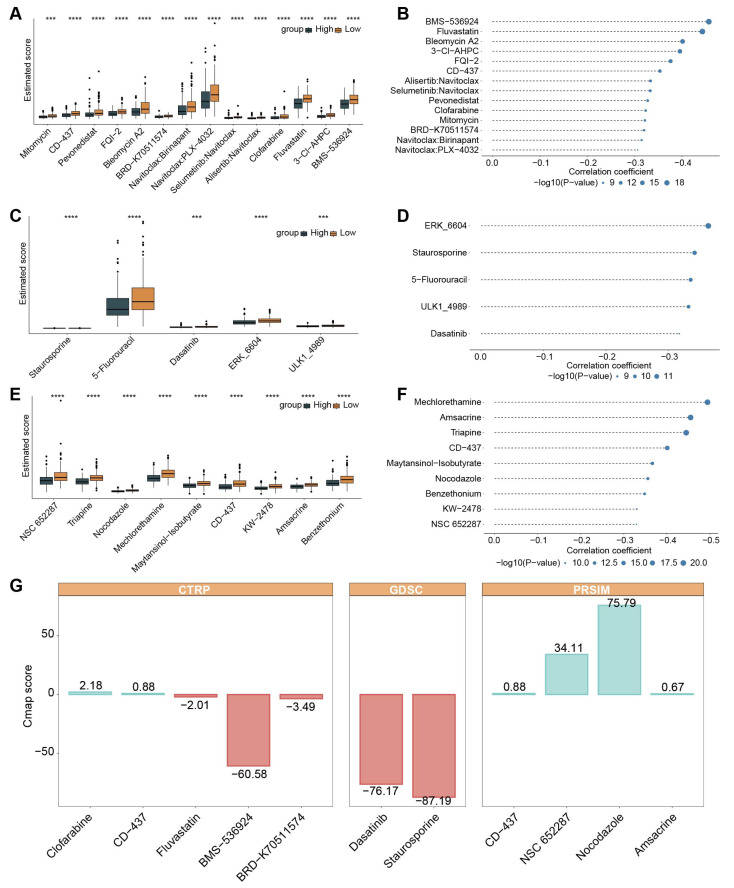
Identifying candidate drugs for patients with the high-MPCDI subtype. (**A**,**B**) Differential drug response analysis and Spearman correlation results for 14 CTRP-derived compounds. (**C**,**D**) Differential drug response analysis and Spearman correlation results for five GDSC-derived compounds. (**E**,**F**) Differential drug response analysis and Spearman correlation results for nine PRISM-derived compounds. (**G**) CMap analysis to identify the most promising candidate drugs for treating patients with the high-MPCDI subtype. **** *p* < 0.0001 and *** *p* < 0.001.

**Figure 8 ijms-26-00118-f008:**
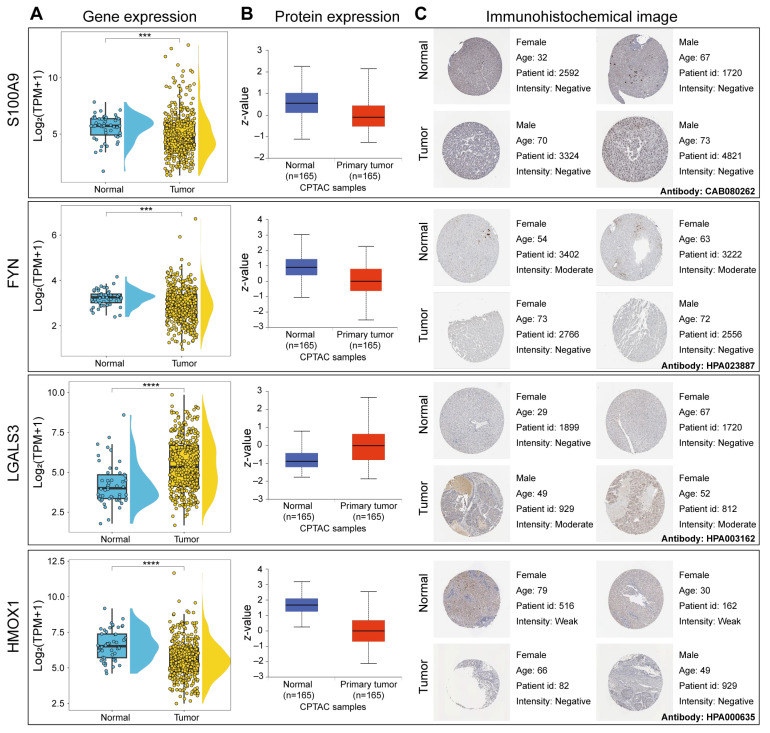
RNA and protein expression levels of MPCDI signature genes. (**A**) RNA expression levels of MPCDI signature genes in the TCGA dataset. (**B**) Protein expression levels of MPCDI signature genes in the CPTAC dataset. (**C**) Representative immunohistochemical images of MPCDI signature genes in normal liver and HCC tissues from the HPA database. **** *p* < 0.0001 and *** *p* < 0.001.

**Table 1 ijms-26-00118-t001:** The information on HCC samples.

Clinical Features	TCGA-LIHC	GSE14520	GSE116174	ICGC-LIRI-JP
**SAMPLE TYPE** ^1^				
Tumor	365	221	64	231
Normal	50	220	−	−
**OS** ^2^				
Alive	234	136	37	189
Dead	131	85	27	42
**AGE** ^3^				
≤60	173	181	46	49
>60	192	40	18	182
**GENDER** ^4^				
Male	246	191	58	170
Female	119	40	18	182
**STAGE** ^5^				
Stage I	170	93	8	36
Stage II	84	77	45	105
Stage III	83	49	11	71
Stage IV	4	0	0	19

***Note:*** ^1^ The type of tissue sample; ^2^ Overall survival status; ^3^ Age of the patient at the time of sample collection; ^4^ The biological sex of the patient; ^5^ Clinical stage of the disease based on TNM classification.

## Data Availability

All data generated or analyzed during this study are included in this published article and its Appendix A files. The prognostic models and related scripts were uploaded to https://github.com/wanglongxing2020/ML_code (accessed on 3 September 2024).

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
