# Peer review of "MPCD Index for Hepatocellular Carcinoma Patients Based on Mitochondrial Function and Cell Death Patterns"

_ijms, 2024, doi:10.3390/ijms26010118_

Round 1
Reviewer 1 Report
Comments and Suggestions for Authors
In this manuscript, LX Wang and coworkers present the outcome of a computational analysis of the activity of mitochondrial-associated genes in hepatocellular carcinoma (HCC). In this study there an attempt to synthesize data from single cell analysis, 4 cohorts of transcriptome (does not seem to generate any ethnic bias with one Asian (JP) and other Caucasians series), there are copy number variation, mutation analyses and methylation analyses i.e. multi-omics with finally a pathological protein validation from Protein Atlas. On the bulk transcriptome, authors applied WGCNA (network co-expression), and deep learning (post tuning with the best combination of algorithms), immune scoring for immune cells infiltration and finally drug prediction. It is thus a very comprehensive study.
The main issue, that is inherent to computational studies is the lack of link with the real biology ie a truly measurable activity of mitochondria ie mitochondrial transcripts produced from mitochondrial genome not from nuclear genes (as it is the case in the manuscript), mitochondrial copy number, respiration and metabolic activity (Seahorse measurements), etc…
What is the rationale to establish a cell death signature from tumors? It is a bit counterintuitive to look for cell death in immortalized cells. Could the authors explain a bit more this apparently paradoxical hypothesis?
In single cells the authors analyzed signatures in 38,311 cells but only143 are hepatocytes ie 0.3% of the total? The single cell analysis has nothing to do with HCC but everything to do with tumor microenvironment. It is the mitochondrial activity of the microenvironmental cells that is therefore important.
Results, line 121: Provide the precision that the 1,136 gene list is the mitocarta list
How it is possible to obtain a 1,711 differentially active gene list? (line 125)
Line 139: the manner the authors obtained the programmed cell death genes (n=1796) in unclear?
Concerning OMICs analysis, it should be said that MSI And KRAS mutation are barely relevant features in HCC.
Figure 4B; it is very difficult to figure out wha tis similar and what is different. Is it possible to propose another figure easier to interprete visually?
Minor issues
Introduction line 79: “they must overcome “ ? Who must overcome ?
Line 235: there are no such things as mRNA mutations. Genomic DNA should be mutated.
Line 247, “HMOX1 and FYN were single nucleotide polymorphisms 247 (SNPs) and were missense mutations”: polymorphisms and mutations although they are looking the same on DNA have different meaning and biological origin. Were they somatic mutations or germline mutations ie most probably SNPs?
Author Response
The main issue, that is inherent to computational studies is the lack of link with the real biology ie a truly measurable activity of mitochondria ie mitochondrial transcripts produced from mitochondrial genome not from nuclear genes (as it is the case in the manuscript), mitochondrial copy number, respiration and metabolic activity (Seahorse measurements), etc…
Answer: Thank you for your question. The mitochondrial gene set in this study was derived from the authoritative mitochondrial function database (MitoCarta3.0), one of the most comprehensive mitochondrial-related gene databases currently available, summarizing mitochondrial-related genes with experimental evidences in humans and other mammals. The 1,136 genes were experimentally verified to be involved in the structure, function, and metabolic activities of mitochondria, such as key functions such as energy metabolism (such as oxidative phosphorylation), metabolic pathways, stress response, and signal transduction. Additionally, we strongly agree with you that direct mitochondrial functional data (such as mitochondrial gene expression, copy number, metabolic activity measurements, etc.) are key parameters for studying mitochondrial function. However, on the one hand, the data and methods used in current research are still mainly based on genetic data from public databases. On the other hand, the gene and protein expression data in public databases are mainly for the nuclear genome, and there are very few expression data resources for the mitochondrial genome. One of the main reasons is that the expression and regulation of the mitochondrial genome are different from those of the nuclear genome, and current data resources are not comprehensive enough. In the future, as the experimental techniques of mitochondrial transcriptomics and metabolomics become more mature and sequencing data become richer, more data support will be provided for research related to mitochondrial function.
What is the rationale to establish a cell death signature from tumors? It is a bit counterintuitive to look for cell death in immortalized cells. Could the authors explain a bit more this apparently paradoxical hypothesis?
Answer: Thank you for your question. Although malignant tumor cells exhibit "immortal" characteristics, it does not mean that they completely escape the cell death process. Instead, it inhibits or selectively activates specific PCD modes by regulating key nodes in the death pathway. The development of malignant tumors is often accompanied by a dynamic balance between cell death and survival. For example, mitochondrial dysfunction may activate or inhibit specific modes of programmed cell death, and the dynamic balance that is critical for tumor aggressiveness and adaptability [1]. Furthermore, studies have shown that in the tumor microenvironment, subpopulation cells may activate certain cell death pathways (such as pyroptosis and ferroptosis) to adapt to harsh environments, such as hypoxia and metabolic stress. These death mechanisms drive the evolution of malignant cell subpopulations [2-3]. Therefore, analysis of tumor cell death signatures can provide insights into the adaptive mechanisms of malignant cells.
There are two main purposes for constructing cell death signatures in this study. (1) Elucidate the relationship between mitochondrial function and PCD and understand its underlying molecular mechanism. (2) Explore new methods to classify and genotype tumors by quantifying cell death signatures to assist in personalized treatment. The evaluation and analysis results of invasiveness, prognosis and treatment sensitivity suggest that the method in this paper has certain application value.
In summary, we believe that it is feasible to explore the characteristics of cell death in immortalized cells, which is of great significance for revealing the biological mechanisms of tumors.
References:
[1] Jin P, Jiang J, Zhou L et al. Mitochondrial adaptation in cancer drug resistance: prevalence, mechanisms, and management. Journal of Hematology & Oncology 2022; 15: 97. (Cited by 110)
[2] Stockwell BR, Angeli JPF, Bayir H et al. Ferroptosis: a regulated cell death nexus linking metabolism, redox biology, and disease. Cell 2017; 171: 273-85. (Cited by 5584)
[3] Fang Y, Tian S, Pan Y et al. Pyroptosis: A new frontier in cancer. Biomedicine & Pharmacotherapy 2020; 121: 109595. (Cited by 855)
NOTE: Citation data are from Google Scholar (as of Dec. 22, 2024).
In single cells the authors analyzed signatures in 38,311 cells but only143 are hepatocytes ie 0.3% of the total? The single cell analysis has nothing to do with HCC but everything to do with tumor microenvironment. It is the mitochondrial activity of the microenvironmental cells that is therefore important.
Answer: Thank you for your question. The single-cell data in this study came from the GSE162616 dataset [1], which contains single-cell sequencing data of tumor tissues and their microenvironment cells from HCC patients. The proportion of hepatocytes is low and proportion of microenvironment cells such as immune cells (such as NK and T cells) is high, which is common in HCC tumor microenvironment studies [1-2]. In HCC single-cell sequencing, hepatocytes are frequently replaced by cancer cells and infiltrating immune cells within the tumor area, so the number of detected hepatocytes is relatively small. Immune cells such as NK cells and T cells in the tumor microenvironment of hepatocellular carcinoma have obvious infiltration characteristics, reflecting the dynamic changes of tumor tissue immunity. Therefore, the phenomenon of a small proportion of hepatocytes and a high proportion of immune cells in this study is not a special case, it is consistent with the data characteristics and the biological characteristics of HCC tumor tissue. Although hepatocytes account for a low proportion, changes in the expression of hepatocyte-related mitochondrial functional genes in each cell type can reflect the overall metabolic characteristics of HCC tissue to a certain extent.
References:
[1] Liu H, Zhao R, Qin R, et al. Panoramic comparison between NK cells in healthy and cancerous liver through single-cell RNA sequencing. Cancer Biology & Medicine 2022; 19: 1334.
[2] Feng Q, Huang Z, Song L, et al. Combining bulk and single-cell RNA-sequencing data to develop an NK cell-related prognostic signature for hepatocellular carcinoma based on an integrated machine learning framework. European Journal of Medical Research 2023; 28: 306.
Results, line 121: Provide the precision that the 1,136 gene list is the mitocarta list
Answer: Thanks for the reminder. The 1,136 mitochondrial-related genes were obtained from the MitoCarta 3.0 database (https://www.broadinstitute.org/mitocarta/mitocarta30-inventory-mammalian-mitochondrial-proteins-and-pathways). In addition, the list of mitochondrial function-related genes is shown in Supplementary Table1.
How it is possible to obtain a 1,711 differentially active gene list? (line 125)
Answer: Thank you for your suggestion. In this study, the FindAllMarkers function was used to perform differential analysis on the two groups of cells and 1,711 differentially active genes were obtained (p < 0.05). And we newly supplemented the list with 1,711 differentially active genes, please see Supplementary Table2 (a new supplementary table).
Line 139: the manner the authors obtained the programmed cell death genes (n=1796) in unclear?
Answer: Thank you for your question. Through literature research [1-3], we collected genes (N = 2,464) related to 20 PCD mechanisms, and after removing duplicates (N = 668), 1,796 genes related to PCD were finally obtained for subsequent analysis. The gene list please see Supplementary Table3.
References:
[1] Qin H, Abulaiti A, Maimaiti A et al. Integrated machine learning survival framework develops a prognostic model based on inter-crosstalk definition of mitochondrial function and cell death patterns in a large multicenter cohort for lower-grade glioma. Journal of Translational Medicine 2023; 21: 588.
[2] Liu X, Nie L, Zhang Y et al. Actin cytoskeleton vulnerability to disulfide stress mediates disulfidptosis. Nature cell biology 2023; 25: 404-14.
[3] Yi X, Li J, Zheng X et al. Construction of PANoptosis signature: Novel target discovery for prostate cancer immunotherapy. Molecular Therapy-Nucleic Acids 2023; 33: 376-90.
Concerning OMICs analysis, it should be said that MSI And KRAS mutation are barely relevant features in HCC.
Answer: Thank you for your suggestion. As pointed out by the reviewer, MSI and KRAS mutation are rare in HCC and therefore were not analyzed in depth as key features in this study. To help readers better understand the research focus, some description was added in Section “4.1 Multi-omics characteristics of HCC subtypes”, “Some features, such as MSI and KRAS mutation, were only weakly associated with HCC, and were therefore not included in subsequent analyses.” Please see Line 602-604.
Figure 4B; it is very difficult to figure out what is similar and what is different. Is it possible to propose another figure easier to interprete visually?
Answer: Thank you for your suggestion. We modified the presentation of Figures 4B and 4C, and labeled the type names in the subfigures to improve readability. Figure 4B shows the copy number variation distribution of the two subtypes, and Figure 4C shows the deletion variation distribution of the two subtypes.
Introduction line 79: “they must overcome “? Who must overcome?
Answer: Thanks to your reminder, they are referring to malignant cells. And we modified the wording, “Malignant cells must overcome various forms of cell death and mitochondrial dysfunction to promote malignant cell growth.” Please see Line 91-93.
Line 235: there are no such things as mRNA mutations. Genomic DNA should be mutated.
Answer: Thank you for your correction. In this study, we did perform CNV and SNV analysis at the genome level.
And the term “mRNA mutations” is indeed inappropriate and has been changed to “gene-level mutations”. Please see Line 261.
Line 247, “HMOX1 and FYN were single nucleotide polymorphisms 247 (SNPs) and were missense mutations”: polymorphisms and mutations although they are looking the same on DNA have different meaning and biological origin. Were they somatic mutations or germline mutations ie most probably SNPs?
Answer: Thank you for your question. We completely agree with what you said about the difference between polymorphisms and mutations. In this study, the maftools annotated HMOX1 and FYN variants as SNPs, and further classified as missense mutations. It should be noted that the annotation of SNPs in the maftools package is based on the variation information of tumor samples. Since this study did not include matched normal samples, it is impossible to clearly distinguish between germline polymorphisms and somatic mutations. This study focused on tumor-specific analysis, so we speculated that they were somatic mutation events. In addition, the FYN mutation also had a frameshift deletion, which also suggested the possibility of somatic mutation.

Reviewer 2 Report
Comments and Suggestions for Authors
This study introduces the Mitochondrial Functional Activity-associated Programmed Cell Death Index (MPCDI), which helps classify HCC patients based on their mitochondrial function and cell death patterns. By utilizing advanced machine learning techniques and analyzing multi-omics data, authors identified a signature of four key genes that serve as independent risk factors for patient prognosis.
Comments:
1- The authors need to expand on the pathophysiology of hepatocellular carcinoma in the introduction section
2- The manuscript is generally well-structured and presents the research findings in a logical manner. However, some sections could benefit from clearer explanations, particularly in the methodology. For instance, a more detailed description of the machine learning algorithms used would enhance the reader's understanding of the analysis process.
3- The integration of scRNA-seq and RNA-seq data is a strong point of the study. However, the authors should elaborate on the challenges faced during data integration and how these were addressed.
4- While the identification of the four prognostic genes (S100A9, FYN, LGALS3, and HMOX1) is noteworthy, the manuscript would benefit from a discussion on the validation of these genes in independent cohorts.
5- The figures and tables are generally well-presented. However, some figures could be improved by including more detailed legends that explain the data being presented. This would aid in the interpretation of the results.
6- A list of abbreviations should be added to the manuscript
Author Response
1-The authors need to expand on the pathophysiology of hepatocellular carcinoma in the introduction section
Answer: Thank you for your suggestion, we have added content related to the pathophysiology of hepatocellular carcinoma. Please see Line 42-55.
The pathophysiology of HCC is multifaceted, driven by a complex interplay of genetic, epigenetic, and environmental factors [1]. The development of HCC is closely associated with a tumor-prone microenvironment characterized by persistent inflammation, extracellular matrix remodeling, and abnormal cellular regeneration. These processes result in genetic instability, the accumulation of somatic mutations, and epigenetic alterations, which collectively promote hepatocarcinogenesis [2]. At the molecular level, HCC is driven by genetic alterations, including mutations in key oncogenes and tumor suppressor genes such as TERT, CTNNB1, and TP53 [3]. Additionally, dysregulation of key signaling pathways, including the Wnt-β-catenin pathway, which is frequently activated in HCC and drives cell proliferation, and the TGF-β signaling pathway, known for its dual role in tumor suppression and progression, plays a critical role in hepatocarcinogenesis. The MAPK and PI3K/AKT pathways are also commonly altered, contributing to enhanced survival, growth, and metabolic reprogramming of HCC cells [4-5].
References:
[1] Torrens L, Puigvehí M, Torres-Martín M et al. Hepatocellular carcinoma in Mongolia delineates unique molecular traits and a mutational signature associated with environmental agents. Clinical Cancer Research 2022; 28: 4509-20. (Cited by 10)
[2] Ozen C, Yildiz G, Dagcan AT et al. Genetics and epigenetics of liver cancer. New biotechnology 2013; 30: 381-4. (Cited by 120)
[3] Totoki Y, Tatsuno K, Covington KR et al. Trans-ancestry mutational landscape of hepatocellular carcinoma genomes. Nature genetics 2014; 46: 1267-73. (Cited by 814)
[4] Huang A, Yang X-R, Chung W-Y et al. Targeted therapy for hepatocellular carcinoma. Signal transduction and targeted therapy 2020; 5: 146. (Cited by 539)
[5] Ng CK, Dazert E, Boldanova T et al. Integrative proteogenomic characterization of hepatocellular carcinoma across etiologies and stages. Nature communications 2022; 13: 2436. (Cited by 85)
NOTE: Citation data are from Google Scholar (as of Dec. 22, 2024).
2-The manuscript is generally well-structured and presents the research findings in a logical manner. However, some sections could benefit from clearer explanations, particularly in the methodology. For instance, a more detailed description of the machine learning algorithms used would enhance the reader's understanding of the analysis process.
Answer: Thank you for your suggestion, we have supplemented the usage and related parameters of ten algorithms. Please see Line 564-580.
The RSF was implemented by the randomForest package with the parameters: ntree = 1000 and nodesize = 5. The Enet, Lasso and Ridge were implemented by the glmnet package, the regularization parameter λ was determined by cross-validation, and the L1-L2 trade-off parameter α was set to 0-1 (interval = 0.1). The stepwise Cox was implemented by the survival package, using the stepwise algorithm of AIC (Akaike information criterion), with the directional modes of the stepwise search set to “both”, “backward”, and “forward”, respectively. The CoxBoost was implemented by the CoxBoost package. The plsRcox was implemented by the plsRcox package, the cv2.plsRcox function was used to determine the required number of components, the plsRcox function was used to fit generalized linear models for partial least squares regression. The SuperPC was implemented by the superpc package, and it was a supervised PCA. The GBM was implemented through the gbm package, cross-validation was used to select the number of decision trees with the smallest cross-validation error, the gbm function was used to fit the generalized boosting regression model. The survive-SVM was implemented through survivalsvm package, this algorithm combined the classification ability of support vector machines and the characteristics of survival analysis.
3-The integration of scRNA-seq and RNA-seq data is a strong point of the study. However, the authors should elaborate on the challenges faced during data integration and how these were addressed.
Answer: Thank you for your support and suggestion, we further elaborate on the current challenges and research ideas of this study in the revised manuscript. Please see Line 91-120.
For example, Qin et al. [1] developed a prognostic model based on inter-crosstalk definition of mitochondrial function and cell death patterns. In their study, only a simple association between genes related to mitochondrial function and PCD-related genes was established at the gene expression level (Pearson co-expression analysis, r > 0.6, p < 0.001). In general, the method is relatively simple and the data format is single.
To fully understand the molecular mechanisms of mitochondrial dysfunction and cell death in tumors, information at the cellular level must be considered. So, our study intends to use scRNA-seq data to mine genes related to mitochondrial function, and on this base, use RNA-seq to calculate the sample score as a key indicator.
Besides, in the integrated use of scRNA-seq data and RNA-seq data, problems such as differences in data sources, batch effects, and technical bias are often encountered [2-3]. In this study, strict data quality control, data standardization conversion, canonical correlation analysis, and other methods were used to integrate different samples, reduce technical deviations between samples, and ensure data consistency.
[1] Qin H, Abulaiti A, Maimaiti A et al. Integrated machine learning survival framework develops a prognostic model based on inter-crosstalk definition of mitochondrial function and cell death patterns in a large multicenter cohort for lower-grade glioma. Journal of Translational Medicine 2023; 21: 588.
[2] Luecken MD, Büttner M, Chaichoompu K et al. Benchmarking atlas-level data integration in single-cell genomics. Nature methods 2022; 19: 41-50. (Cited by 787)
[3] Conesa A, Madrigal P, Tarazona S et al. A survey of best practices for RNA-seq data analysis. Genome biology 2016; 17: 1-19. (Cited by 3291)
4-While the identification of the four prognostic genes (S100A9, FYN, LGALS3, and HMOX1) is noteworthy, the manuscript would benefit from a discussion on the validation of these genes in independent cohorts.
Answer: Thank you for your question, we also believe that validation in genetically independent cohorts is very important. This study utilized all available data resources as much as possible and performed multi-level validation on these four genes. However, independent cohorts of both tumor and normal samples are currently very limited in public datasets. For example, GSE116174 and ICGC-LIRI-JP lack normal sample controls. Even some cohorts that include both tumor and normal samples, such as GSE76427, do not fully cover the four genes. Only the GSE14520 dataset meets the requirements, but this dataset was published in 2010. Then we calculated the expression distribution of these four genes in the GSE14520 dataset, and the gene expression trends were consistent with expectations. Corresponding supplements have also been made in the manuscript. Please see Line 385-387 and Figure S6 (a new figure).
We believe that as the HCC data set continues to be enriched and improved in the future, there will be more and more sufficient evidence to support it.
5-The figures and tables are generally well-presented. However, some figures could be improved by including more detailed legends that explain the data being presented. This would aid in the interpretation of the results.
Answer: Thank you for your suggestion, we have revised Figure 1, Figure 4, Figure S1 and Graphical Abstract.
6-A list of abbreviations should be added to the manuscript
Answer: Thank you for your suggestion. The list of abbreviations has been added to the manuscript.
ACD: Accidental Cell Death
CIBERSORT: Cell-type Identification by Estimating Relative Subpopulations of RNA Transcripts
CNV: Copy Number Variation
CTRP: Cancer Therapeutics Response Portal
DCA: Decision Curve Analysis
Enet: Elastic Net
ESTIMATE: Estimation of STromal and Immune Cells in MAlignant Tumor Tissues using Expression Data
FDR: False Discovery Rate
GBM: Generalized Boosted Model
GDSC: Genomics of Drug Sensitivity in Cancer
GEO: Gene Expression Omnibus
GSEA: Gene Set Enrichment Analysis
GSVA: Gene Set Variation Analysis
HCC: Hepatocellular Carcinoma
HPA: Human Protein Atlas
IC50: The half-maximal inhibitory concentration
ICGC: International Cancer Genome Consortium
k-NN: k-Nearest Neighbors
